# Associations of Dynapenic Abdominal Obesity and Frailty Progression: Evidence from Two Nationwide Cohorts

**DOI:** 10.3390/nu16040518

**Published:** 2024-02-13

**Authors:** Boran Sun, Jingyue Wang, Yanbo Wang, Wenbo Xiao, Yang Liu, Yuan Wang, Yongjie Chen, Wenli Lu

**Affiliations:** Department of Epidemiology and Statistics, School of Public Health, Tianjin Medical University, Tianjin 300070, China; sunboran65@tmu.edu.cn (B.S.); 13820803139@163.com (J.W.); 18812555122@163.com (Y.W.); xiaowenbo199703@163.com (W.X.); liuyang123@tmu.edu.cn (Y.L.); wangyuan@tmu.edu.cn (Y.W.); chenyongjie@tmu.edu.cn (Y.C.)

**Keywords:** frailty, dynapenia, abdominal obesity, transitions, middle-aged and older adults

## Abstract

The associations of dynapenic abdominal obesity and transitions with frailty progression remain unclear among middle-aged and older adults. We included 6937 participants from the China Health and Retirement Longitudinal Study (CHARLS) and 3735 from the English Longitudinal Study of Aging (ELSA). Participants were divided into non-dynapenia and non-abdominal obesity (ND/NAO), abdominal obesity alone (AO), dynapenia alone (D), and dynapenic abdominal obesity (D/AO). Frailty status was assessed by the frailty index (FI), and a linear mixed-effect model was employed to analyze the associations of D, AO, D/AO, and transitions with frailty progression. Participants with AO, D, and D/AO had increased FI progression compared with ND/NAO in both cohorts. D/AO possessed the greatest additional annual FI increase of 0.383 (95% CI: 0.152 to 0.614), followed by D and AO in the CHARLS. Participants with D in the ELSA had the greatest magnitude of accelerated FI progression. Participants who transitioned from ND/NAO to D and from AO to D/AO presented accelerated FI progression in the CHARLS and ELSA. In conclusion, dynapenic abdominal obesity, especially for D/AO and D, presented accelerated frailty progression. Our findings highlighted the essential intervention targets of dynapenia and abdominal obesity for the prevention of frailty progression.

## 1. Introduction

Frailty, characterized by a decreased physiological reserve and losses of function in multidimensional systems, has become a global public health concern [1]. Frailty is associated with adverse outcomes such as disability, falls, depression, and mortality [2,3,4]. Potential reversible risk factors should be identified to delay frailty progression, as a dynamic nature of frailty has been reported in the recent literature [5,6,7,8].

Musculoskeletal problems, characterized by muscle weakness and low muscle mass, are a common condition among middle-aged and older adults [9,10]. However, muscle mass and muscle strength do not show a synchronous change with age. Recent studies have reported that skeletal muscle strength declined faster than muscle mass [11] and that muscle atrophy accounted for only a small fraction of the decline in muscle strength [12]. Therefore, it is necessary to make a clear differentiation between them. The term dynapenia was thus proposed as a distinct condition of age-associated loss of skeletal muscle strength. Weakness, with underlying mechanisms of impairments in the structure and function of the nervous and muscular system [13], has been considered to be a key contributor to physical frailty [14]. Moreover, the proportion of fat mass, especially abdominal fat, increases with age and has become a public health concern [15]. Abdominal obesity has been identified as a crucial risk determinant of frailty among older adults [16] and, compared with general obesity indicators such as body mass index (BMI), abdominal obesity indicators such as waist circumference (WC), and waist-to-height ratio (WHR), have a better performance in frailty prediction [17,18].

Although the evidence showed adverse impacts of dynapenia and abdominal obesity on frailty, mounting evidence suggested that there might be an interaction between dynapenia and abdominal obesity, reciprocally regulated by the dysfunction of skeletal muscle and adipose tissue [19]. On the one hand, dynapenia resulted in weight gain due to an impaired ability to perform physical activities [20]. On the other hand, fat in muscle and around the abdomen accumulates with advanced age, which predisposes the individual to decreased muscle strength through inflammatory and endocrine regulation pathways [21]. Therefore, dynapenic abdominal obesity, defined as the concurrence of dynapenia and abdominal obesity, was associated with greater adverse outcomes such as low gait speed, falls, function disability, and mortality [22,23,24]. However, the independent and combination effects of dynapenia and abdominal obesity on frailty progression among middle-aged and older adults remain unclear. Furthermore, given that the status of dynapenic abdominal obesity changes over time, the effects of dynapenic abdominal obesity transitions on frailty progression also remain to be elucidated.

Thus, the present study aimed to investigate the relationship between dynapenic abdominal obesity status and its transitions with frailty trajectory among Chinese and English middle-aged and older adults using two national longitudinal cohorts to explore potential targets for the prevention of frailty progression in the context of aging.

## 2. Materials and Methods

### 2.1. Sample

In this study, our data were derived from two ongoing national representative cohorts, the China Health and Retirement Longitudinal Study (CHARLS) and the English Longitudinal Study of Aging (ELSA). Detailed descriptions of these two cohorts can be found in publications elsewhere [25,26]. In this study, we analyzed data from wave 1 to wave 4 in the CHARLS and wave 2 to wave 7 in the ELSA. The use of the CHARLS and the ELSA was approved by the ethics committees of Peking University (IRB00001052-11015) and London Multicenter Research (MREC/01/2/91), respectively. All participants provided written informed consent.

Participants who were aged ≥ 50 years and who attended the baseline interview were included. The exclusion criteria, as referred to in the previous study [27], were set to ensure statistical power without introducing systematic bias and were defined as follows: (1) missing or invalid data regarding WC and handgrip strength; (2) those without a frailty index (FI) reassessment and >10% missing FI items at baseline and follow-up. Finally, 6937 and 3735 participants from the CHARLS and ELSA, respectively, were included as our analytical sample. Among them, 5418 and 3390 were reidentified as having dynapenic abdominal obesity status and were included in the transition analyses.

### 2.2. Assessment of Dynapenia and Abdominal Obesity

Handgrip strength was a valid indicator to identify dynapenia participants [24]. In the CHARLS, handgrip strength was measured using a handgrip dynamometer (YuejianTM WL-1000, Nantong Yuejian Physical Measurement Instrument Co., Ltd., Nantong, China); it was measured twice in the dominant hand, and the maximal value was used [28]. In the ELSA, handgrip strength was measured using a hand dynamometer (Smedley; range: 0~100 kg) using the dominant hand with three maximum trials, and the highest value was considered in the analysis [29]. Dynapenia was defined as a handgrip strength of <28 kg for men and of <18 kg for women in the CHARLS, according to the Asian Working Group for Sarcopenia 2019 (AWGS 2019) [30], and of <26 kg for men and of <16 kg for women in the ELSA based on the previous study [31].

Abdominal obesity was assessed by waist circumference according to the country-specific criteria. Waist circumference was measured twice using a flexible and non-elastic tape at the midpoint between the iliac crest and last rib [32]. Abdominal obesity was defined as a waist circumference of ≥90 cm for men and of ≥85 cm for women in the CHARLS [33], and of >102 cm for men and of >88 cm for women in the ELSA [31].

Participants were divided into four groups based on their dynapenia and abdominal obesity status: non-dynapenia and non-abdominal obesity (ND/NAO); only abdominal obesity (AO); only dynapenia (D) and dynapenic abdominal obesity (D/AO).

### 2.3. Evaluation of Frailty

The frailty index, characterized by the accumulation of age-related health deficits, was recommended as an effective instrument to evaluate frailty status among older adults [34]. A total of 32 common and comparative items, including domains of medical conditions (13 items), ADL/IADL disabilities (10 items), functional abilities (7 items), cognition, and depression in the CHARLS and ELSA, were selected to construct the FI based on the previous study [27]. Specifically, the value of 1 indicated the presence of health deficits except for the cognition item. Cognition scores ranged from 0 to 1, and a higher value presented a worse cognition level. The 32-FI, ranging from 0 to 100, was calculated as the sum of health deficits divided by 32 and multiplied by 100. Individuals with a higher 32-FI value had a higher degree of frailty, and frailty status was defined as 32-FI ≥ 25. The median value of the corresponding item was used to impute missing FI items [35]. The detailed descriptions can be seen in Appendix A.

### 2.4. Covariates

The covariates in this study included age, sex, education level, marital status, smoking, drinking, and BMI, which were related to frailty and comparable among the two cohorts. Similar to the previous study [27], covariates were divided for the consistency between the CHARLS and ELSA. Education level was divided into less than lower secondary, upper secondary and vocational training, and tertiary according to the 1997 International Standard Classification of Education scale (ISCED-97) [36]. Marital status was classified as married or partnered and other marital status. Smoking was classified into current, former, and never smokers. Drinking was categorized as at least once a month, <1 time per month, and never drinkers based on drinking frequency in the past year. Weight (kg) and height (m) were measured using a scale and stadiometer, and BMI was then calculated.

### 2.5. Statistical Analysis

One-way ANOVA or Kruskal Wallis rank sum tests were used to compare the baseline continuous variables across ND/NAO, AO, D, and D/AO groups when appropriate. A chi-square test was performed to compare the categorical characteristics across groups.

Linear mixed-effect models were employed to analyze the associations of dynapenic abdominal obesity with FI progression. The model specification and assumption for the normality of residuals were checked using a Q-Q plot to meet the statistical requirements. Dynapenic abdominal obesity group, follow-up time (years since baseline), the interaction of group and time, and covariates were included as the independent variables for fixed effects. Outcome variables were repeated measurements of FI, and ND/NAO was considered to be the reference. Random effects for the intercept and slope (time) were taken into account for the individual FI differences at baseline and different rates of FI change during follow-up, based on the previous study [27]. In the linear mixed-effect models, the coefficients of group and time denoted the average baseline FI difference as compared with the reference and the overall FI annual change rate, respectively. The regression coefficient of interaction between group and time indicated additional annual FI changes, which stood for the accelerated frailty progression in comparison with the reference. Further stratified analyses were conducted by sex and age. The likelihood ratio test was used to examine the interaction effects between group, time, and stratified factors. The relationship between dynapenic abdominal obesity transitions and FI progression was analyzed with similar methods, taking stable ND/NAO as the reference. Non-response household and individual weights and person-level nurse interview weights were adjusted in the CHARLS and ELSA, respectively. Sensitivity analyses were performed to test the robustness of results and explore the potential causes of heterogeneity: (1) explore whether D alone, AO alone, and obesity measured with BMI ≥ 25 kg/m^2^ in the CHARLS and ≥30 kg/m^2^ in the ELSA could change the main results; (2) analyze the associations of dynapenic abdominal obesity with FI progression across different domains. All statistical analyses were performed using SAS 9.4 (SAS Institute Inc., Cary, NC, USA.) and R software (Version 4.2.2). The false discovery rate (FDR) method was used to adjust *p* values when conducting multiple comparisons. Otherwise, two-sided *p* values < 0.05 were considered statistically significant.

## 3. Results

### 3.1. Baseline Characteristics of Participants

With a median follow-up of 7.0 and 9.7 years, 6937 and 3735 participants were included in the CHARLS and ELSA, respectively (Figure 1). The mean age was 60.9 (SD = 7.3) and 63.7 (SD = 8.0), and the analytic samples were both composed predominantly of women. In the CHARLS, 2630 (37.9%), 499 (7.2%), and 265 (3.8%) participants were divided into AO, D, and D/AO. Compared with ND/NAO, individuals with D/AO were older, less married or partnered, less educated, and had a lower proportion of smoking and drinking. In the ELSA, only 3.1% of participants were D/AO, and they were older and less educated. The FI and frailty prevalence was the highest among D/AO, followed by D, AO, and ND/NAO in both cohorts (Appendix A). Pairwise comparisons of baseline characteristics showed a lower socioeconomic level (i.e., lower education level and less married or partnered) among D and D/AO in both cohorts. Participants with D/AO were less likely to be current smokers and drink at least once a month (Appendix A). Compared with excluded individuals due to loss to follow-up, the analytic samples were more likely to be younger, married, not current smokers, and have a lower FI (Appendix A). The detailed baseline FI in the two cohorts was presented in Appendix A.

### 3.2. Associations of Dynapenic Abdominal Obesity with Frailty at Baseline

In the CHARLS, although significantly higher baseline FI was not found among AO (*β*: −0.225, 95% CI: −0.933 to 0.484, *p* = 0.534), D/AO (*β*: 5.265, 95% CI: 3.889 to 6.640, *p* < 0.001) and D (*β*: 4.043, 95% CI: 2.995 to 5.090, *p* < 0.001) had a higher baseline FI, compared with ND/NAO. In the ELSA, a significantly higher baseline FI was shown among D/AO (*β*: 14.942, 95% CI: 12.679 to 17.204, *p* < 0.001), D (*β*: 11.984, 95% CI: 9.668 to 14.300, *p* < 0.001), and AO (*β*: 1.768, 95% CI: 0.786 to 2.749, *p* < 0.001) (Table 1).

### 3.3. Associations of Dynapenic Abdominal Obesity with Frailty Progression

Individuals with AO, D, and D/AO had an increased FI progression in comparison with ND/NAO in both cohorts (Table 2). D/AO had the greatest additional annual FI increase of 0.383 (95% CI: 0.152 to 0.614, *p* = 0.001), followed by D (*β*: 0.368, 95% CI: 0.188 to 0.549, *p* < 0.001), and AO (*β*: 0.214, 95% CI: 0.120 to 0.308, *p* < 0.001) in the CHARLS. D had the greatest magnitude of additional annual FI increase (*β*: 0.479, 95% CI: 0.254 to 0.704, *p* < 0.001), then followed by D/AO (*β*: 0.273, 95% CI: 0.065 to 0.480, *p* = 0.010), and AO (*β*: 0.169, 95% CI: 0.098 to 0.241, *p* < 0.001) in the ELSA. After excluding frail participants at baseline, the associations between D, AO, and D/AO and accelerated FI progression were stronger, and the greatest FI progression still existed among D/AO and D in the CHARLS and ELSA, respectively (Table 3).

Sex difference for the associations of dynapenic abdominal obesity and FI progression was found in the CHARLS (*p* for interaction = 0.032). D/AO and D were dominant in the increased FI progression among males and females, respectively (Appendix A). The age group modified the associations in both cohorts (all *p* for interaction < 0.001). In the CHARLS, the accelerated effect of D/AO on FI progression increased with age, and D/AO had the greatest impact on FI progression among 60–70 years old individuals. In the ELSA, D, and D/AO accelerated FI progression among 70+ years old individuals, with D being the greatest (Appendix A).

### 3.4. Associations of Dynapenic Abdominal Obesity Transitions with Frailty Progression

After 4 years of follow-up, 1.9% and 1.6% of ND/NAO transitioned to D/AO in the CHARLS and ELSA, respectively. On the contrary, 6.3% and 8.0% of D/AO moved to ND/NAO, respectively (Appendix A). Compared with stable ND/NAO, participants who transitioned from ND/NAO to D (*β*: 0.388, 95% CI: 0.119 to 0.657, *p* = 0.005) presented an accelerated FI progression in the CHARLS. In addition, participants moved from AO to D/AO (*β*: 0.469, 95% CI: 0.206 to 0.731, *p* < 0.001) accelerated FI progression in the ELSA (Figure 2). ND/NAO to D and AO to D/AO presented accelerated FI progression for the intergroup comparisons (Appendix A).

### 3.5. Sensitivity Analysis

In line with the main analysis, AO alone and D alone still accelerated FI progression in both cohorts, and D had a greater effect (Appendix A). Additionally, we did not find an accelerated FI progression for D/O (Appendix A), which corroborated the necessity of utilizing D/AO to monitor the FI progression. Second, we found that D was dominant in accelerating the progression of ADL/IADL disabilities, and D/AO had the greatest effect on increasing the progressions of medical conditions and functional abilities in the CHARLS. In the ELSA, D was predominant in accelerating the progressions of medical conditions, disabilities, and functional abilities (Appendix A).

## 4. Discussion

Evidence of the associations between dynapenic abdominal obesity and its transitions and frailty progression is rare and less representative, especially for the Chinese. As we know, this is the first study to examine the role of dynapenic abdominal obesity and its transitions on frailty progression. In the present study, we found that AO, D, and D/AO accelerated the frailty progression. D/AO and D showed the greatest magnitude of increasing frailty progression among Chinese and English middle-aged and older adults, respectively. Transitions from ND/NAO to D and from AO to D/AO presented the accelerated progression of frailty.

A main finding of the present study was that abdominal and dynapenia phenotypes, AO and D, had independent and combined synergic influences on accelerating the progression of frailty. Older adults with D/AO and D should be considered to be the key population for further frailty intervention. Dynapenia and abdominal obesity were both contemporary public health concerns among older adults. Previous evidence showed that decreased muscle strength contributed to physical frailty. Meanwhile, dynapenia was considered to be a key characteristic of sarcopenia, which was found to be related to frailty and its transitions [37,38]. A recent meta-analysis also revealed that a set of metabolic, hematologic, and inflammatory biomarkers were shared by frailty and sarcopenia [39]. Obesity and frailty share analogous pathophysiological pathways, including oxidative stress, systemic inflammation, and insulin resistance [40]. Abdominal obesity was regarded as an indicator of metabolic disorders. The increased level of insulin resistance would accelerate the replacement of skeletal muscle with adipose tissue, leading to decreased muscle strength and the occurrence of frailty [41]. Our findings were consistent with previous evidence and highlighted the essential roles of improving muscle health, healthy body size, and metabolic conditions in preventing frailty progression. Moreover, the potential interaction between muscle strength and adiposity has been revealed in mounting studies. Aging leads to frailty characterized by numerous physical and mental health deficits and might be a rational explanation for the associations of dynapenic and abdominal obesity and frailty progression. Dynapenia was presented by changes in muscle mass and architecture, fiber replacement, and defects in motor neurons [14,20]. Age-related neurophysiological changes, including reduced activation in motor neurons and decreased muscle quantity and quality, led to the occurrence of dynapenia [42]. Meanwhile, age-related abdominal fat accumulation disturbed the process of motor neuron repairment and muscle anabolism under the increased oxidative stress and low-grade inflammatory level [21], which further degraded muscle fibers and accelerated the progression of dynapenia [43,44]. To the best of our knowledge, it was the first study to provide dynamic sights on the adverse impact of dynapenic abdominal obesity transitions on frailty aging. He and his colleagues found that transitions from metabolic healthy status to metabolic unhealthy status accelerated the frailty progression, irrespective of obesity status for Chinese and English middle-aged and older adults [27]. Similar to He’s study, another main finding in this study was that transitions to dynapenia accelerated the progression of frailty in comparison with stable ND/NAO, regardless of abdominal obesity status. This study indicated feasible means to delay or prevent frailty progression in the aging process, which was keeping muscle strength and preventing muscle aging.

In this study, we hypothesized that D/AO had the greatest effect on accelerating the frailty progression in both cohorts. However, the magnitude of accelerated FI progression was heterogeneous among AO, D, and D/AO in the CHARLS and ELSA. First, although CHARLS and ELSA were sister cohorts with similar designs, there were still differences between the two cohorts in the aspect of race, follow-up duration, and lifestyle behaviors, which might explain the inconsistent results to some extent [25,26]. At present, China and the UK are in different aging processes. Compared with the UK, the low level of elderly-adaptation renovations and social welfare among Chinese older adults might lead to less active attitudes and behaviors to cope with aging. Second, we found that several differences existed in the distributions of dynapenic abdominal obesity status and baseline FI between the two cohorts. A lower prevalence of AO and higher baseline FI for ND/NAO in the CHARLS and a lower proportion of D in the ELSA might contribute to the heterogeneity. Third, another plausible reason was the inconsistent effects of dynapenic abdominal obesity phenotypes on FI progression across FI domains. D/AO had the greatest influence on the progression of medical conditions and functional abilities, and D was dominant in increasing the progression of disabilities in the CHARLS. Yang and his colleagues found that dynapenic obesity was associated with a greater risk of ADL/IADL disabilities in comparison with dynapenia or obesity alone among older Chinese adults [45]. Our study used the representative Chinese cohort and additionally revealed that dynapenia would lead to a greater progression of ADL/IADL disabilities. Although there was heterogeneity among our results, dynapenia, and abdominal obesity should be regarded as two potential targets for frailty management. Further studies with larger sample sizes and longer follow-up durations are needed to validate our findings and explore underlying mechanisms to effectively guide the public to cope with frailty in the context of aging.

This study had several strengths. First, we used two prospective cohorts with strict implementation procedures and nationwide samples, therefore elucidating that our results were nationally representative. Second, we considered the dynapenic abdominal obesity status at baseline and transitions during follow-up, which was limited in previous investigations but crucial for the prevention of frailty progression. Third, diverse sensitivity analyses ensured the robustness of our results. However, our study also had some limitations. First, the definition of dynapenic abdominal obesity had no uniform criteria. For this issue, we used the most common definition in previous studies. In addition, we replaced abdominal obesity with general obesity measured by BMI and found no significantly accelerated frailty progression for dynapenic obesity, which corroborated the rationality of the definition used in the main analyses. Additionally, the heterogeneity regarding participants in the CHARLS and ELSA limited the generalizability of our results. Further studies in multiracial populations and different clinical settings should be conducted to validate our findings. Second, 27.6% and 48.3% of participants from the CHARLS and ELSA were excluded due to loss of follow-up. They were more likely to be under a higher degree of frailty at baseline and had an increased progression of frailty compared with the analytic samples, which may lead to selection bias and underestimating the association strength. Third, the FI used in this study was a validated instrument reflecting comorbidity conditions, which might overestimate the frailty prevalence due to the great impact of abdominal obesity on numerous chronic diseases. There is an urgent need to find appropriate frailty assessment tools that are not affected by weakness and abdominal obesity. Lastly, although we had adjusted for multiple covariates, there might still be additional residual or unmeasured confounders, such as dietary factors and genetic susceptibility. Potential reverse causality might also mislead our results, and causal conclusions should not be drawn. Further meta-analyses are essential to clarify the associations to guide frailty management from the perspectives of adiposity and muscle.

Despite the conclusions that should be applied with caution when exploring potential intervention strategies, multidisciplinary teams, including healthcare providers, nutritionists, therapists, as well as policymakers, should develop comprehensive strategies to prevent and manage dynapenia and abdominal obesity against frailty among older adults. Health education should be widely carried out in primary healthcare institutions to improve the awareness of maintaining muscle strength and preventing abdominal obesity from middle age. Furthermore, public health programs, including aerobic exercises, resistance training, and nutritional supplements, should be developed to improve muscle function, reduce abdominal fat, and prevent frailty progression.

## 5. Conclusions

AO, D, and D/AO accelerated the frailty progression among Chinese and English middle-aged and older adults. D/AO and D presented the greatest magnitude of accelerating the frailty progression in the CHARLS and ELSA, respectively. Regardless of abdominal obesity status, transitions to dynapenia accelerated the frailty progression. Our findings highlighted the essential intervention targets of dynapenia and abdominal obesity for the prevention of frailty progression.

## Figures and Tables

**Figure 1 nutrients-16-00518-f001:**
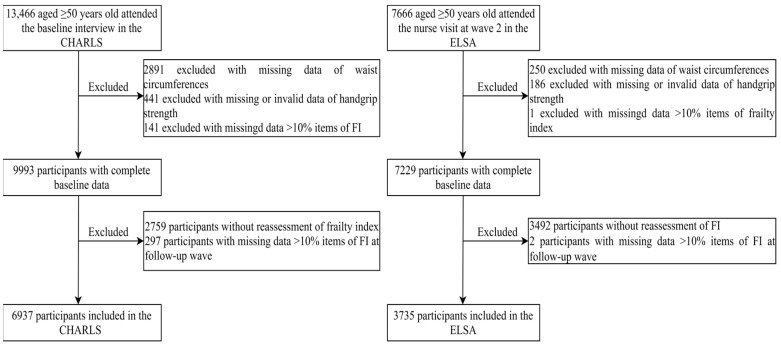
Flowchart of analytic samples.

**Figure 2 nutrients-16-00518-f002:**
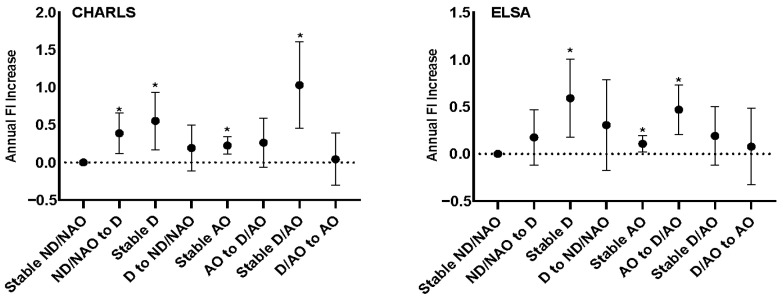
Relationship between dynapenic abdominal obesity transitions with frailty progression. * denotes *p* < 0.05.

**Table 1 nutrients-16-00518-t001:** Associations of dynapenic abdominal obesity with the baseline FI.

Status	CHARLS			ELSA		
	*β*	95% CI	*p*	*β*	95% CI	*p*
ND/NAO	Ref	-	-	Ref	-	-
AO	−0.225	−0.933 to 0.484	0.534	1.768	0.786 to 2.749	<0.001
D	4.043	2.995 to 5.090	<0.001	11.984	9.668 to 14.300	<0.001
D/AO	5.265	3.889 to 6.640	<0.001	14.942	12.679 to 17.204	<0.001

**Table 2 nutrients-16-00518-t002:** Associations of dynapenic abdominal obesity with FI progression.

Variables	CHARLS			ELSA		
	*β*	95% CI	*p*	*β*	95% CI	*p*
Time, years	0.772	0.709 to 0.834	<0.001	0.444	0.394 to 0.495	<0.001
ND/NAO	Ref	-	-	Ref	-	-
AO	−0.048	−0.728 to 0.631	0.889	1.397	0.470 to 2.324	0.003
D	3.733	2.734 to 4.732	<0.001	11.068	8.833 to 13.303	<0.001
D/AO	5.142	3.780 to 6.504	<0.001	13.046	10.882 to 15.211	<0.001
ND/NAO × time	Ref	-	-	Ref	-	-
AO × time	0.214	0.120 to 0.308	<0.001	0.169	0.098 to 0.241	<0.001
D × time	0.368	0.188 to 0.549	<0.001	0.479	0.254 to 0.704	<0.001
D/AO × time	0.383	0.152 to 0.614	0.001	0.273	0.065 to 0.480	0.010

**Table 3 nutrients-16-00518-t003:** Associations of dynapenic abdominal obesity with FI progression among non-frail participants at baseline.

Variables	CHARLS			ELSA		
	*β*	95% CI	*p*	*β*	95% CI	*p*
Time, years	0.839	0.773 to 0.899	<0.001	0.461	0.411 to 0.511	<0.001
ND/NAO	Ref	-	-	Ref	-	-
AO	−0.279	−0.789 to 0.232	0.285	1.094	0.179 to 1.435	0.012
D	1.524	0.725 to 2.324	<0.001	4.483	2.782 to 6.184	<0.001
D/AO	1.096	−0.062 to 2.254	0.064	4.519	2.534 to 6.505	<0.001
ND/NAO × time	Ref	-	-	Ref	-	-
AO × time	0.220	0.122 to 0.317	<0.001	0.201	0.129 to 0.274	<0.001
D × time	0.519	0.322 to 0.716	<0.001	0.727	0.469 to 0.985	<0.001
D/AO × time	0.755	0.480 to 1.030	<0.001	0.686	0.388 to 0.985	<0.001

## Data Availability

The datasets analyzed during the current study are available at https://charls.pku.edu.cn/ (accessed on 29 October 2022) and https://beta.ukdataservice.ac.uk/ (accessed on 14 September 2023), respectively.

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
