# Peer review of "Associations of Dynapenic Abdominal Obesity and Frailty Progression: Evidence from Two Nationwide Cohorts"

_nutrients, 2024, doi:10.3390/nu16040518_

Round 1

Reviewer 1 Report

Comments and Suggestions for Authors

The study you've outlined suggests that the combination of dynapenia (loss of muscle strength associated with aging) and abdominal obesity is associated with accelerated frailty progression in middle-aged and older adults. Here are some suggestions for the study:

1. Further Investigation into Mechanisms:

   - Explore the underlying mechanisms that link dynapenia and abdominal obesity to frailty progression. Investigate whether there are specific biological pathways, hormonal changes, or inflammatory processes that contribute to this association.

2. Longitudinal Analysis:

   - Consider extending the follow-up period in future studies to capture a longer-term perspective on frailty progression. This could provide more insights into the dynamics of these conditions over time.

3. Subgroup Analysis:

   - Conduct subgroup analyses to explore whether there are specific demographic, lifestyle, or health-related factors that modify the relationship between dynapenia, abdominal obesity, and frailty progression. This could help identify vulnerable populations.

4. Intervention Studies:

   - Design intervention studies to assess whether targeted interventions for dynapenia and abdominal obesity can mitigate frailty progression. Investigate the effectiveness of exercise programs, nutritional interventions, or other lifestyle modifications.

5. Health Education and Promotion:

   - Develop health education programs aimed at raising awareness about the importance of maintaining muscle strength and preventing abdominal obesity in middle-aged and older adults. Emphasize the potential impact on frailty progression.

6. Public Health Implications:

   - Discuss the public health implications of the findings. Consider how the prevention and management of dynapenia, potentially through physical activity programs, and abdominal obesity could be incorporated into public health strategies for aging populations.

7. Clinical Guidelines:

   - If the results support it, consider proposing or updating clinical guidelines for healthcare professionals to include recommendations for assessing and managing dynapenia and abdominal obesity as potential risk factors for frailty.

8. Multidisciplinary Approach:

   - Encourage a multidisciplinary approach involving healthcare providers, nutritionists, physical therapists, and policymakers to develop comprehensive strategies for preventing and managing dynapenia and abdominal obesity in older adults.

9. Replication Studies:

   - Encourage replication studies in different populations and settings to validate the generalizability of the findings.

10. Communication and Implementation:

    - Develop clear communication strategies for disseminating the study's findings to healthcare professionals, policymakers, and the general public. Highlight the practical implications and actions that can be taken based on the results.

Comments on the Quality of English Language

The study you've outlined suggests that the combination of dynapenia (loss of muscle strength associated with aging) and abdominal obesity is associated with accelerated frailty progression in middle-aged and older adults. Here are some suggestions for the study:

1. Further Investigation into Mechanisms:

   - Explore the underlying mechanisms that link dynapenia and abdominal obesity to frailty progression. Investigate whether there are specific biological pathways, hormonal changes, or inflammatory processes that contribute to this association.

2. Longitudinal Analysis:

   - Consider extending the follow-up period in future studies to capture a longer-term perspective on frailty progression. This could provide more insights into the dynamics of these conditions over time.

3. Subgroup Analysis:

   - Conduct subgroup analyses to explore whether there are specific demographic, lifestyle, or health-related factors that modify the relationship between dynapenia, abdominal obesity, and frailty progression. This could help identify vulnerable populations.

4. Intervention Studies:

   - Design intervention studies to assess whether targeted interventions for dynapenia and abdominal obesity can mitigate frailty progression. Investigate the effectiveness of exercise programs, nutritional interventions, or other lifestyle modifications.

5. Health Education and Promotion:

   - Develop health education programs aimed at raising awareness about the importance of maintaining muscle strength and preventing abdominal obesity in middle-aged and older adults. Emphasize the potential impact on frailty progression.

6. Public Health Implications:

   - Discuss the public health implications of the findings. Consider how the prevention and management of dynapenia, potentially through physical activity programs, and abdominal obesity could be incorporated into public health strategies for aging populations.

7. Clinical Guidelines:

   - If the results support it, consider proposing or updating clinical guidelines for healthcare professionals to include recommendations for assessing and managing dynapenia and abdominal obesity as potential risk factors for frailty.

8. Multidisciplinary Approach:

   - Encourage a multidisciplinary approach involving healthcare providers, nutritionists, physical therapists, and policymakers to develop comprehensive strategies for preventing and managing dynapenia and abdominal obesity in older adults.

9. Replication Studies:

   - Encourage replication studies in different populations and settings to validate the generalizability of the findings.

10. Communication and Implementation:

    - Develop clear communication strategies for disseminating the study's findings to healthcare professionals, policymakers, and the general public. Highlight the practical implications and actions that can be taken based on the results.

.Those are just suggestions

Author Response

The study you've outlined suggests that the combination of dynapenia (loss of muscle strength associated with aging) and abdominal obesity is associated with accelerated frailty progression in middle-aged and older adults. Here are some suggestions for the study:

1.      Further Investigation into Mechanisms:

Explore the underlying mechanisms that link dynapenia and abdominal obesity to frailty progression. Investigate whether there are specific biological pathways, hormonal changes, or inflammatory processes that contribute to this association. 

Responses: Thank you for your suggestions. Potential biological pathways had been added in our revised manuscripts as follows: “Aging leads to frailty presented by numerous physical and mental health deficits and might be a rational explanation for the associations of interaction between dynapenia and abdominal obesity and frailty progression. Previous evidence showed that dynapenia was presented by changes in muscle mass and architecture, fibers replacement and defects in motor neurons. Age-related neurophysiological changes, including reduced activation in motor neuron, muscle quantity and quality could result in the occurrence of dynapenia. Meanwhile, age-related abdominal fat accumulation disturbed the repairment process of motor neurons and muscle anabolism under the increased oxidative stress and low-grade inflammatory level, which further degraded muscle fibers and accelerated the progression of dynapenia. ”

2.      Longitudinal Analysis:

Consider extending the follow-up period in future studies to capture a longer-term perspective on frailty progression. This could provide more insights into the dynamics of these conditions over time.

Responses: Thank you for your suggestions. Further studies with larger sample size and longer follow-up duration are needed to validate our finding and explore underlying mechanisms.

3.      Subgroup Analysis

Conduct subgroup analyses to explore whether there are specific demographic, lifestyle, or health-related factors that modify the relationship between dynapenia, abdominal obesity, and frailty progression. This could help identify vulnerable populations.

Responses: Thank you for your suggestions. Subgroup analyses for sex and age were conducted to identify the vulnerable populations. The detailed results had been revised in the manuscript as follows: “Sex difference for the association between dynapenic abdominal obesity and frailty progression was found in the CHARLS (p for interaction=0.032). D/AO and D was dominant on increased FI progression among male and female (Supplemental Table S8). Age group modified the association in both cohorts (all p for interaction<0.001). In the CHARLS, the accelerated effect of D/AO on FI progression increased with age and D/AO had the greatest impact on FI progression among 60-70 years old individuals. In the ELSA, D and D/AO had accelerated FI progression among 70+ years old individuals, with D being the greatest (Supplemental Table S9). ” Therefore, older adults with D/AO and D phenotype might be considered as the key population for further frailty intervention.

4.      Intervention Studies 

Design intervention studies to assess whether targeted interventions for dynapenia and abdominal obesity can mitigate frailty progression. Investigate the effectiveness of exercise programs, nutritional interventions, or other lifestyle modifications.

Responses: Thank you for your suggestions. Further intervention studies should be designed to determine the effectiveness and optimal means of intervention, such as nutritional supplementation, exercising, and lifestyle education.

5.      Health Education and Promotion:

Develop health education programs aimed at raising awareness about the importance of maintaining muscle strength and preventing abdominal obesity in middle-aged and older adults. Emphasize the potential impact on frailty progression.

Responses: Thank you for your suggestions. We had added the relevant suggestions in the revised manuscript as follows: “Health education should be widely carried out in primary health units to improve the awareness of maintaining muscle strength and preventing abdominal obesity from middle age.”

6.      Public Health Implications 

Discuss the public health implications of the findings. Consider how the prevention and management of dynapenia, potentially through physical activity programs, and abdominal obesity could be incorporated into public health strategies for aging populations. 

Responses: Thank you for your suggestions. Dynapenia and abdominal obesity are two common geriatric conditions, which have significant impacts on frailty progression. Efficient strategies, such as aerobic and resistance training, as well as nutritional supplements, should be developed for improving muscle function, reducing abdominal fat, and preventing frailty progression. The revised texts were as follows: “Furthermore, public health programs including aerobic and resistance training, and nutritional supplements should be developed aiming at improving muscle function, reducing abdominal fat, and preventing frailty progression”

7.      Clinical Guidelines:

If the results support it, consider proposing or updating clinical guidelines for healthcare professionals to include recommendations for assessing and managing dynapenia and abdominal obesity as potential risk factors for frailty.

Responses: Thank you for your suggestions. In the present study, heterogeneity regarding participants in the CHARLS and ELSA limited the generalizability of our results. Further studies in multiracial populations and different clinical settings are urgently needed to propose recommendations and clinical guidelines for frailty among older adults.

8.      Multidisciplinary Approach

Encourage a multidisciplinary approach involving healthcare providers, nutritionists, physical therapists, and policymakers to develop comprehensive strategies for preventing and managing dynapenia and abdominal obesity in older adults.

Responses: Thank you for your suggestions. We had added the relevant suggestions in the revised manuscript as follows: “Multidisciplinary teams including healthcare providers, nutritionists, therapists, as well as policy makers should develop comprehensive strategies to prevent and manage dynapenia and abdominal obesity against frailty among older adults.”

9.      Replication Studies:

Encourage replication studies in different populations and settings to validate the generalizability of the findings.

Responses: Thank you for your suggestions. We had added the relevant suggestions in the revised manuscript as follows: “Further studies in multiracial populations and different clinical settings would be conducted to validate our findings.”

10.   Communication and Implementation

Develop clear communication strategies for disseminating the study's findings to healthcare professionals, policymakers, and the general public. Highlight the practical implications and actions that can be taken based on the results.

Responses: Thank you for your suggestions. The suggestions you provide above are beneficial to our further researches. Relevant content had been added to the revised manuscript as follows: “Despite the conclusions in this study need to be taken with caution when exploring potential intervention strategy, multidisciplinary teams including healthcare providers, nutritionists, therapists, as well as policy makers should develop comprehensive strategies to prevent and manage dynapenia and abdominal obesity against frailty among older adults. Health education should be widely carried out in primary health units to improve the awareness of maintaining muscle strength and preventing abdominal obesity from middle age. Furthermore, public health programs including aerobic and resistance training, and nutritional supplements should be developed aiming at improving muscle function, reducing abdominal fat, and preventing frailty progression. ”

Reviewer 2 Report

Comments and Suggestions for Authors

This is a very interesting and important topic. I would like to know what does this manuscript add relating to the previous publication: 

J Nutr Health Aging   2023;27(12):1188-1195.  doi: 10.1007/s12603-023-2039-1.

Dynapenic Abdominal Obesity as a Risk Factor for Metabolic Syndrome in Individual 50 Years of Age or Older: English Longitudinal Study of Ageing

P C Ramírez 1R de Oliveira MáximoD Capra de OliveiraA F de SouzaM Marques LuizM L Bicigo DelinocenteA SteptoeC de OliveiraT da Silva Alexandre    

Author Response

This is a very interesting and important topic. I would like to know what does this manuscript add relating to the previous publication: J Nutr Health Aging. 2023;27(12):1188-1195.  doi: 10.1007/s12603-023-2039-1. “Dynapenic Abdominal Obesity as a Risk Factor for Metabolic Syndrome in Individual 50 Years of Age or Older: English Longitudinal Study of Ageing”.

Responses: Thank you for your interest. Dynapenic abdominal obesity was reported to increase the risk of MetS, compared to abdominal obesity alone in the previous study. The synergic action of dynapenic abdominal obesity could guide specific clinical strategies to prevent the metabolic changes. The outcome of present study was frailty, which was measured with 32-item accumulated health deficits. Mets can only reflect the health status from the aspect of metabolism, but lacks the comprehensive investigation of physical and mental health for the middle-aged and older adults. The present study revealed that dynapenia, abdominal obesity, and their combinations contributed to the accelerated frailty progression, compared to the non-dynapenia and non-abdominal obesity. In addition, our study added additional understanding of association between dynapenic abdominal obesity status transitions and frailty progression among older adults, so as to better guide the prevention and management of frailty from the perspective of abdominal obesity and dynapenia.

Reviewer 3 Report

Comments and Suggestions for Authors

1. Sample Selection (Lines 64-78):

• The study used data from two large prospective cohorts, which is generally a strength. Authors should write that the exclusion criteria (e.g., missing data on waist circumference (WC), handgrip strength, or frailty index (FI) items) do not introduce systematic bias.

• The study included a substantial number of participants, which is sufficient for statistical power. However, it has been verified that the participants were representative of a broader population to ensure generalizability.

2. Assessment of Dynapenia and Abdominal Obesity (Lines 79-93):

• The use of handgrip strength as an indicator of dynapenia and waist circumference for abdominal obesity is standard and appropriate. Do we have data on cut-off values for different races?

3. Evaluation of Frailty (Lines 98-109):

• The use of the 32-item frailty index is a comprehensive approach. Each item is appropriately weighted, and the accumulation of deficits approach is suitable for the objectives of the study. What about the validation of these methods in different races and ages?

4. Covariates (Lines 110-119):

• This study considered a range of relevant covariates, which is important for controlling potential confounding factors. Did the authors consider all other potential confounders present in the data of two cohorts and that the categorization of variables such as education level, marital status, smoking, and drinking is appropriate and justified?

5. Statistical Analysis (Lines 120-142):

• The use of one-way ANOVA, Kruskal-Wallis, and Chi-square tests for baseline comparisons is standard. However, the assumptions of these tests were met (e.g., normality for ANOVA).

• The use of a linear mixed model to analyze FI progression is appropriate for longitudinal data. Ensure that the model is correctly specified, including the random effects for intercept and slope, and that the assumptions of the model are checked and met (e.g., normality of residuals).

• Adjustment for non-response and person-level weights is crucial for reducing bias; these weights are correctly calculated and applied.

• It is important that all p-values are interpreted in the context of the study design and that multiple comparison corrections are considered if multiple tests are conducted.

6. Have the authors consulted biostatistics to apply the statistical methods used?

The methods were not given a detailed assessment that might be necessary to identify specific statistical issues or methodological flaws. The authors provided a general review of the literature. They should provide assumptions of the statistical tests and models used to ensure that the interpretations and conclusions are fully supported by the results.

7. Baseline Characteristics of Participants (Lines 145-159):

• This study provides a detailed description of the baseline characteristics. Are comparisons between groups (AO, D, D/AO, and ND/NAO) statistically and clinically significant? A description of the results is provided in the manuscript.

• The exclusion of certain samples due to loss to follow-up and their impact on the representativeness of the analytic sample should be carefully considered. The authors should discuss the potential for bias and how it might have affected the results.

8. Association of Dynapenic Abdominal Obesity with Frailty at Baseline (Lines 163-173):

• Differences in the associations between CHARLS and ELSA should be explained. If substantial differences exist, the reasons for these differences should be explored and discussed.

9. Association between Dynapenic Abdominal Obesity and Frailty Progression (Lines 174-202):

• The progression of frailty index (FI) was analyzed in different groups. The interpretation of the coefficients and the significance levels should be clear, indicating how much FI progresses for each group compared to the reference (ND/NAO).

• Interaction effects (e.g., sex and age group) were explored, which is a strength as it shows the variability of the associations across different subgroups with proper conduction and cautious interpretation. Are the results presented and interpreter?

• The use of confidence intervals and significance levels provides insights into the precision and statistical significance of the findings. However, we should also discuss the clinical significance of these findings.

10. Association of Dynapenic Abdominal Obesity Transitions with Frailty Progression (Lines 203-221):

• Analysis of transitions between different states (e.g., ND/NAO to D/AO) and their association with FI progression is a significant addition. Is the potential for reverse causality or confounding addressed?

• I assume that the sensitivity analyses conducted to test the robustness of the results should be clearer and adequately explained and that their findings support the main results of the study.

11. Contextualization of Findings (Lines 223-278):

• The potential interaction between muscle strength and adiposity and its implications for frailty progression are well-articulated. Has any alternative explanation or mechanism been considered and discussed: aging leads to frailty with limitations and enhances fat accumulation?

• Heterogeneity in the findings between the CHARLS and ELSA cohorts is acknowledged and discussed. It is important to explore and present the potential reasons for these differences, and how they might affect the generalizability of the findings.

A discussion of the dynamic impact of dynapenia and abdominal obesity on frailty is valuable. However, the conclusions drawn are cautious and reflect the limitations of the study, particularly when discussing potential interventions or preventive measures.

12. Limitations of the Study (Lines 285-297):

• The authors acknowledge the lack of uniform criteria for defining dynapenic abdominal obesity, the potential for selection bias due to loss to follow-up, the limitations of using the frailty index (FI) in the context of abdominal obesity, and the possibility of residual or unmeasured confounders. It is important to discuss these limitations in detail, including their potential impact on the findings, and how they might be addressed in future research.

Comments on the Quality of English Language

Good, minor misspelling 

Author Response

1. Sample Selection (Lines 64-78)

 The study used data from two large prospective cohorts, which is generally a strength. Authors should write that the exclusion criteria (e.g., missing data on waist circumference (WC), handgrip strength, or frailty index (FI) items) do not introduce systematic bias.

The study included a substantial number of participants, which is sufficient for statistical power. However, it has been verified that the participants were representative of a broader population to ensure generalizability.

Responses: Thank you for your suggestions. The CHARLS and ELSA study was both representative national cohorts for Chinese and England middle-aged and older community-dwelling adults. Two cohorts with different ethnics were utilized to analyze the association of dynapenia and abdominal obesity and frailty progression. Therefore, our results were limited to Chinese and England middle-aged and older adults and we had added this limitation to the revised manuscript. In addition, Similar with He’s study (10.1002/jcsm.13169), participants with eligible and complete exposure factors, namely dynapenia and abdominal obesity, were included. Moreover, individuals without frailty index (FI) reassessment and >10% missing FI items at baseline and follow-up was excluded to explore the association between dynapenia and abdominal obesity status transitions and frailty progression under the premise of ensuring the statistical power. Thus, the exclusion criteria did not introduce systematic bias as reported in the previous study. The revised sentences were as follows in the sample selection: “The exclusion criteria were referred to previous study to ensure statistical power without introducing systematic bias and defined as follows.”

2. Assessment of Dynapenia and Abdominal Obesity (Lines 79-93):

The use of handgrip strength as an indicator of dynapenia and waist circumference for abdominal obesity is standard and appropriate. Do we have data on cut-off values for different races?

Responses: Thank you for your suggestions. Since the definition of dynapenic abdominal obesity status had no uniform criteria up till now, we used the most common definition. Dynapenia measured by handgrip strength was assessed by the Asian Working Group for Sarcopenia 2019 criteria and previous ELSA evidence. The cut-off values were <28 kg for men and <18 kg for women for Chinese, and <26 kg for men and <16 kg for England residents. The cut-off values for abdominal obesity were based on country-specific criteria and previous studies. The revised sentences were as follows: “Dynapenia was defined as handgrip strength <28 kg for men and <18 kg for women in the CHARLS according to the Asian Working Group for Sarcopenia 2019 (AWGS 2019), and <26 kg for men and <16 kg for women in the ELSA; Abdominal obesity was defined by waist circumference ≥90 cm for men and ≥85 cm for women in the CHARLS, and >102 cm for men and >88 cm for women in the ELSA.”

3. Evaluation of Frailty (Lines 98-109):

The use of the 32-item frailty index is a comprehensive approach. Each item is appropriately weighted, and the accumulation of deficits approach is suitable for the objectives of the study. What about the validation of these methods in different races and ages?

Responses: Thank you for your suggestions. Frailty was evaluated by the frailty index (FI), which was calculated as the accumulation of age-related health deficits. He and his colleagues constructed the FI following standard procedures as described previously (10.1100/tsw.2001.58).  As reported in He’s study (10.1002/jcsm.13169), 32-item frailty index had been used to assess the relationships between metabolic heterogeneity of obesity with frailty progression among adults aged no less than 50 years old in the CHARLS and ELSA, thus validating its effectiveness among middle-aged and older adults in these two cohorts.

4. Covariates (Lines 110-119):

This study considered a range of relevant covariates, which is important for controlling potential confounding factors. Did the authors consider all other potential confounders present in the data of two cohorts and that the categorization of variables such as education level, marital status, smoking, and drinking is appropriate and justified?

Responses: Thank you for your suggestions. The outcome of frailty index is a multidimensional indicator including domains of medical conditions, ADL/IADL disability, functional ability, cognition, and depression in this study. We adjusted for basic demographic and lifestyle factors to obtain accurate estimation. Although there are large number of variables in two cohorts, all the covariates included in our study were comparable and essential to adjust for analyzing the association between dynapenic abdominal obesity status and frailty progression. For the consistency of covariates classification between CHARLS and ELSA, we revised in the manuscript as follows: “Similar with previous study, essential covariates were included and divided for the consistency between CHARLS and ELSA.”

5. Statistical Analysis (Lines 120-142):

  • The use of one-way ANOVA, Kruskal-Wallis, and Chi-square tests for baseline comparisons is standard. However, the assumptions of these tests were met (e.g., normality for ANOVA).

Responses: Thank you for your suggestions. After statistical testing, the assumptions of these tests were met, such as the normality. And we corrected the statement in the Statistical Analysis section as follows: “One-way ANOVA or Kruskal Wallis rank sum tests was used for comparing the baseline continuous variables across the ND/NAO, AO, D, and D/AO group as appropriate. Chi-square test was performed to compare the categorical characteristics across groups.”

  • The use of a linear mixed model to analyze FI progression is appropriate for longitudinal data. Ensure that the model is correctly specified, including the random effects for intercept and slope, and that the assumptions of the model are checked and met (e.g., normality of residuals).

Responses: Thank you for your suggestions. As reported in previous study, linear-mixed effects model was used to analyze the relationship between dynapenic abdominal obesity and trajectories of ADL (10.1093/gerona/gly182), IADL (10.1016/j.clnu.2017.09.018) , and long-term gait speed (10.1093/ageing/afab093) using ELSA longitudinal cohorts. Model specification and the assumptions for the normality of residuals were checked to ensure the accuracy of our results. We clarified the use of linear mixed-effect model in the statistical analysis section as follows: “The model specification and assumptions for normality of residuals were checked using Q-Q plot to meet the statistical requirements. Dynapenic abdominal obesity group, follow-up time (years since baseline), the interaction of group and time, and covariates were included as the independent variables for fixed effects. Outcome variables were repeated measurements of FI and ND/NAO was considered as the reference. Similar with previous study [34], random effects for intercept and slope (time) were took into account for the individual FI differences at baseline and different rates of FI change during follow-up. In the linear mixed-effect models, the coefficients of group and time denoted the average baseline FI difference as compared with the reference and the overall FI annual change rate, respectively. The regression coefficient of interaction terms between group and time indicated additional annual FI changes, which stood for accelerated frailty progression in comparison with the reference.”

  • Adjustment for non-response and person-level weights is crucial for reducing bias; these weights are correctly calculated and applied.

Responses: Thank you for your suggestions. These weights were calculated and then provided by the team of CHARLS and ELSA when the dataset was released. Although previous study reported that weight adjustment did not materially alter the result in the CHARLS (10.1093/gerona/glz185), we still utilized provided weights to obtain more accurate and comparative results between CHARLS and ELSA.

  • It is important that all p-values are interpreted in the context of the study design and that multiple comparison corrections are considered if multiple tests are conducted.

Responses: Thank you for your suggestions. In the present study, multiple comparisons were only performed across baseline characteristics for ND/AO, D, AO, and D/AO groups. all P values were adjusted by the false discovery rate (FDR) method when conducting multiple comparisons. We revised the statement in the statistical analysis section as follows: “The false discovery rate (FDR) method was used to adjust p values when conducting multiple comparisons. Otherwise, two-sided P values <0.05 were considered statistically significant.”

6. Have the authors consulted biostatistics to apply the statistical methods used? The methods were not given a detailed assessment that might be necessary to identify specific statistical issues or methodological flaws. The authors provided a general review of the literature. They should provide assumptions of the statistical tests and models used to ensure that the interpretations and conclusions are fully supported by the results.

Responses: Thank you for your suggestions. The linear mixed-effects model had been extensively used as an effective method to analyze the longitudinal trajectory of adverse outcomes, such as ADL, IADL, and gait speed. We clarified the use of linear mixed-effect model in the statistical analysis section as follows: “Linear mixed-effect models were employed to analyze the associations of dynapenic abdominal obesity with FI progression. The model specification and as-sumptions for normality of residuals were checked using Q-Q plot to meet the statistical requirements.”

7. Baseline Characteristics of Participants (Lines 145-159):

  • This study provides a detailed description of the baseline characteristics. Are comparisons between groups (AO, D, D/AO, and ND/NAO) statistically and clinically significant? A description of the results is provided in the manuscript.

Responses: Thank you for your suggestions. We had revised and added the comparisons of baseline characteristics between groups in the Baseline characteristics of participants as follows:” Pairwise comparisons of baseline characteristics showed a lower socioeconomic level (i.e., lower education level and less married or partnered) among D and D/AO group in both cohorts. Participants with D/AO were less likely to be current smoker and drinking at least once a month (Supplemental Table S3 and Table S4).

  • The exclusion of certain samples due to loss to follow-up and their impact on the representativeness of the analytic sample should be carefully considered. The authors should discuss the potential for bias and how it might have affected the results.

Responses: Thank you for your suggestions. We had considered the impact of excluding certain sample due to loss to follow-up in the limitations section of the revised manuscript. “Secondly, 27.6% and 48.3% participants from the CHARLS and ELSA were excluded due to loss to follow-up. They were more likely to be under worse frailty status at baseline and have increased frailty progression compared with analytic sample, which may lead to selection bias and underestimate the association strength. ”

8. Association of Dynapenic Abdominal Obesity with Frailty at Baseline (Lines 163-173):

  • Differences in the associations between CHARLS and ELSA should be explained. If substantial differences exist, the reasons for these differences should be explored and discussed.

Responses: Thank you for your suggestions. In this section, we found that although no significant baseline FI difference between AO and ND/NAO was found in the CHARLS, D and D/AO in both cohorts had a significantly higher baseline FI compared with ND/NAO. Meanwhile, the combination of dynapenia and abdominal obesity had the highest baseline FI in both cohorts. Since the basic trends of dynapenic abdominal obesity and baseline FI were consistent between two cohorts and the main outcome in the present study was the FI progression, we did not explain too much in the discussion section. And the heterogeneity could be attribute to sample characteristics between these cohorts, such as the lower prevalence of AO, the higher baseline FI for the ND/NAO, and different distributions of covariates in the CHARLS. We had corrected in the revised manuscript as follows: “In the CHARLS, although no significant baseline FI difference was found among AO (β: -0.225, 95% CI: -0.933 to 0.484, P=0.534), D/AO (β: 5.265, 95% CI: 3.889 to 6.640, P<0.001) and D (β: 4.043, 95% CI: 2.995 to 5.090, P<0.001) had higher baseline FI, compared with ND/NAO. In the ELSA, significantly higher baseline FI was shown among D/AO (β: 14.942, 95% CI: 12.679 to 17.204, P<0.001), D (β: 11.984, 95% CI: 9.668 to 14.300, P<0.001), and AO (β: 1.768, 95% CI: 0.786 to 2.749, P<0.001)”.

9. Association between Dynapenic Abdominal Obesity and Frailty Progression (Lines 174-202):

  • The progression of frailty index (FI) was analyzed in different groups. The interpretation of the coefficients and the significance levels should be clear, indicating how much FI progresses for each group compared to the reference (ND/NAO).

Responses: Thank you for your suggestions. We clearly described the increased FI progression in each group compared with the ND/NAO group in the revised manuscript as follows: “Individuals with AO, D, and D/AO had increased FI progression in comparison with ND/NAO in both cohorts (Table 2). D/AO had the greatest additional annual FI increase of 0.383 (95% CI: 0.152 to 0.614, P=0.001), followed by D (β: 0.368, 95% CI: 0.188 to 0.549, P < 0.001), and AO (β: 0.214, 95% CI: 0.120 to 0.308, P<0.001) in the CHARLS. D had the greatest magnitude of additional annual FI increase (β: 0.479, 95% CI: 0.254 to 0.704, P<0.001), then followed by D/AO (β: 0.273, 95% CI: 0.065 to 0.480, P=0.010), and AO (β: 0.169, 95% CI: 0.098 to 0.241, P<0.001) compared with ND/NAO in the ELSA. After excluding frail participants at baseline, the associations between D, AO, and D/AO and accelerated FI progression were stronger and the greatest FI progression still existed among D/AO and D in the CHARLS and ELSA, respectively (Table 3)”.

  • Interaction effects (e.g., sex and age group) were explored, which is a strength as it shows the variability of the associations across different subgroups with proper conduction and cautious interpretation. Are the results presented and interpreter?

Responses: Thank you for your suggestions. We had clarified the interaction effects in the section of revised manuscript as follows: “Sex difference for the association between dynapenic abdominal obesity and frail-ty progression was found significant in the CHARLS (p for interaction=0.032). D/AO and D was dominant on increased FI progression among male and female (Supplemental Table S8). Age group modified the association in both cohorts (all p for interaction<0.001). In the CHARLS, the accelerated effect of D/AO on FI progression increased with age and D/AO had the greatest impact among 60-70 years old individuals. In the ELSA, D and D/AO had accelerated FI progression among 70+ years old individuals, with D being the greatest (Supplemental Table S9)”.

  • The use of confidence intervals and significance levels provides insights into the precision and statistical significance of the findings. However, we should also discuss the clinical significance of these findings.

Responses: Thank you for your suggestions. We recombined and reorganized the  descriptions to clarify the statistical and clinical associations between dynapenia abdominal obesity and FI progression as follows: “D/AO had the greatest additional annual FI increase of 0.383 (95% CI: 0.152 to 0.614, P=0.001), followed by D (β: 0.368, 95% CI: 0.188 to 0.549, P < 0.001), and AO (β: 0.214, 95% CI: 0.120 to 0.308, P<0.001) in the CHARLS. D had the greatest magnitude of additional annual FI increase (β: 0.479, 95% CI: 0.254 to 0.704, P<0.001), then followed by D/AO (β: 0.273, 95% CI: 0.065 to 0.480, P=0.010), and AO (β: 0.169, 95% CI: 0.098 to 0.241, P<0.001) compared with ND/NAO in the ELSA.”

10. Association of Dynapenic Abdominal Obesity Transitions with Frailty Progression (Lines 203-221):

  • Analysis of transitions between different states (e.g., ND/NAO to D/AO) and their association with FI progression is a significant addition. Is the potential for reverse causality or confounding addressed?

Responses: Thank you for your suggestions. In the present study, we used the baseline and resurvey data in the two national cohorts to define the transitions and analyze the associations between transitions with FI progression. Prospective cohort studies can provide relatively high-level evidences with a nature of time sequence. Nevertheless, we did not consider the potential reversal confounders when analyzing the mean progression of 32-item frailty index with dynapenic abdominal obesity status transitions. We had added to the limitation section in the revised manuscript as follows: “Potential reversal confounding might mislead our results and causal conclusions could not be drawn. Further meta-analyses are essential to clarify the association so as to guide the frailty management in the perspective of adiposity and muscle”.

  • I assume that the sensitivity analyses conducted to test the robustness of the results should be clearer and adequately explained and that their findings support the main results of the study.

Responses: Thank you for your suggestions. In the present study, Sensitivity analyses were performed to test the robustness of results and explore the causes of heterogeneity: (1) explore whether D and AO alone, and using obesity meas-ured with BMI ≥ 25 kg/m2 in the CHARLS and ≥ 30 kg/m2 in the ELSA could change the main results; (2) analyze the associations of dynapenia and abdominal obesity with FI progression from different domains. And we corrected the expressions to clarify our sensitivity analysis in the revised manuscript as follows: “In line with the main analysis, AO and D alone still accelerated FI progression in both cohorts and D had a greater effect (Supplemental Table S12). Additionally, we did not find accelerated FI progression for D/O (Supplemental Table S13), which corroborated the necessity of utilizing D/AO to monitor the FI progression. Secondly, we found that D was dominant on the accelerated ADL/IADL disability progression and D/AO had the greatest acceleration of medical conditions and functional ability progression in the CHARLS. In the ELSA, D was being predominant on accelerated progression of medical conditions, disability, and functional abilities (Supplemental Table S14). ”

11. Contextualization of Findings (Lines 223-278):

  • The potential interaction between muscle strength and adiposity and its implications for frailty progression are well-articulated. Has any alternative explanation or mechanism been considered and discussed: aging leads to frailty with limitations and enhances fat accumulation?

Responses: Thank you for your suggestions. We had considered aging as a potential explanation of  the association between interaction between muscle strength and adiposity and frailty progression. And we revised in the manuscript as follows: “ Aging leads to frailty presented by numerous physical and mental health deficits and might be a rational explanation for the associations of interaction between dynapenia and abdominal obesity and frailty progression. Previous evidence showed that dynapenia was presented by changes in muscle mass and architecture, fibers replacement and defects in motor neurons. Age-related neurophysiological changes, including reduced activation in motor neuron, muscle quantity and quality could result in the occurrence of dynapenia. Meanwhile, age-related abdominal fat accumulation disturbed the repairment process of motor neurons and muscle anabolism under the increased oxidative stress and low-grade inflammatory level, which further degraded muscle fibers and accelerated the progression of dynapenia.”

  • Heterogeneity in the findings between the CHARLS and ELSA cohorts is acknowledged and discussed. It is important to explore and present the potential reasons for these differences, and how they might affect the generalizability of the findings.

Responses:  Thank you for your suggestions. The evidence to explore the heterogeneity of our results was rare. We further stated the possible causes of heterogeneity in the revised manuscript as follows: “Firstly, although CHARLS and ELSA were sister cohorts with similar designs, there was still differences between these two cohorts in the aspect of race, follow-up duration and lifestyle behaviors, which could explain the inconsistent results to some extents. China and the UK are in different aging processes. Compared with the UK, the current low level of elderly- adaptation renovation and social welfare for the Chinese older adults might lead to less active attitudes and behaviors to cope with aging. Secondly, we found that several differences existed in the distribution of dynapenic abdominal obesity status and baseline FI between two cohorts. A lower prevalence of AO and higher baseline FI for the ND/NAO group in the CHARLS, and lower proportion of D in the ELSA might contribute to the heterogeneity. Thirdly, another plausible reason was that the inconsistent effect of dynapenic abdominal obesity phenotypes on FI progression across FI domains. D/AO had the greatest influence on the progression of medical conditions and functional abilities, and D was dominant in the disability progression in the CHARLS. Yang and his colleagues found that dynapenic obesity was associated with a greater ADL/IADL disability risk in comparison with dynapenia or obesity alone among Chinese older adults. Our study used the representative Chinese cohort and additionally revealed that dynapenia should lead to greater ADL/IADL disability progression. Although there was heterogeneity among our results, dynapenia and abdominal obesity should be regarded as two potential intervention targets for frailty management. Further studies with larger sample size and longer follow-up duration are needed to validate our findings and explore underlying mechanisms, so as to effectively guide the public to cope with frailty in the context of aging.”

  • A discussion of the dynamic impact of dynapenia and abdominal obesity on frailty is valuable. However, the conclusions drawn are cautious and reflect the limitations of the study, particularly when discussing potential interventions or preventive measures.

Responses: Thank you for your suggestions. Although there was heterogeneity for results between the two cohorts, our results still revealed that dynapenic abdominal obesity status played a critical role on preventing the frailty progression. The conclusions in this study need to be taken with caution when exploring potential intervention strategy. We revised the manuscript as follows: “Despite the conclusions in this study need to be taken with caution when exploring potential intervention strategy, multidisciplinary teams including healthcare providers, nutritionists, therapists, as well as policy makers should develop comprehensive strategies to prevent and manage dynapenia and abdominal obesity against frailty among older adults. Health education should be widely carried out in primary health units to improve the awareness of maintaining muscle strength and preventing abdominal obesity from middle age. Furthermore, public health programs including aerobic and resistance training, and nutritional supplements should be developed aiming at improving muscle function, reducing abdominal fat, and preventing frailty progression. ”

12. Limitations of the Study (Lines 285-297):

  • The authors acknowledge the lack of uniform criteria for defining dynapenic abdominal obesity, the potential for selection bias due to loss to follow-up, the limitations of using the frailty index (FI) in the context of abdominal obesity, and the possibility of residual or unmeasured confounders. It is important to discuss these limitations in detail, including their potential impact on the findings, and how they might be addressed in future research.

Responses: Thank you for your suggestions. We had discussed the limitations about their potential impact on the findings and implications for further researches. The revised sentences in the limitation section were as follows: “Firstly, the definition of dynapenic abdominal obesity status had no uniform criteria. For this issue, we used the most common definition in previous researches. In addition, we replaced abdominal obesity with general obesity measured by BMI and found no significant accelerated frailty progression with combination of dynapenia and obesity, which corroborated the rationality of definition used in the main analyses. Additionally, heterogeneity regarding participants in the CHARLS and ELSA limited the generalizability of our results. Further studies in multiracial populations and different clinical settings would be conducted to validate our findings. Secondly, 27.6% and 48.3% participants from the CHARLS and ELSA were excluded due to loss to follow-up. They were more likely to be under worse frailty status at baseline and have increased frailty progression compared with analytic sample, which may lead to selection bias and underestimate the association strength. Thirdly, the FI used in this study was a validated instrument reflecting comorbidities and it might overestimate the frailty prevalence in the context of abdominal obesity due to the great impact of abdominal obesity on numerous chronic diseases.  There is an urgent need to find appropriate frailty assessment tools, which are not affected by weakness and abdominal obesity. Lastly, although we had adjusted for multiple covariates, there might still be additional residual or unmeasured confounders, such as dietary factors and genetic susceptibility. Potential reversal confounding might mislead our results and causal conclusions could not be drawn. Further meta-analyses were essential to clarify the association so as to guide the frailty management in the perspective of adiposity and muscle.”

Reviewer 4 Report

Comments and Suggestions for Authors

It is a privilege to review the manuscript dedicated to the issue of frailty and its determinants in two large cohorts. The following remarks describe issues affecting the quality of the presentation.

1. English editing is required to make the text more comprehensible. Parts of the text that require special attention are the following: 

-  First sentence of the abstract "Little was known..." is not clear

- Present and past tenses should be adjusted

- Introduction: the term "intimate connection" is used to describe relationships between people, other adjectives should be used to underline connection between scientific measures.

- Sentences beginning with the word "While" should be reconstructed to improve the clarity: lines: 55, 178, 208, 217, 266

Lines 91-93 - This part is no clear, because there is no sentence, lack of verb.

The use of plural form "researches" seems improper. Usually the single form "research" is used. Maybe "research studies" could substitute "researches"

2. Abstract - First sentence not clear (see language comments above)

3. Introduction - Please check if citation [13] is adequately used. In the manuscript it relates to dynapenia and physical frailty while the title of the publication [13] is related to cognition.

4. Materials and methods: Clearly described. No comments except for English editing.

5. Results: Good presentation with a lot of supplementary materials with detailed results. One issue that requires clarification: Was low BMI analyzed as a possible frailty correlate? Loss of weight and low BMI are known risk factors for frailty. If it was not included in the study, it should be mentioned as a limitation of the research.

6. Discussion: This section is rather concise, but includes all relevant findings of the study. Expanded discussion on the potential application of the research in practice (clinical medical practice, public health strategies) would enrich the discussion.

7. Conclusions are in line with the results.

8. References: Please, check [13] - is the citation appropriate in the Introduction section?

Comments on the Quality of English Language

As stated above:

English editing is required to make the text more comprehensible. Parts of the text that require special attention are the following: 

-  First sentence of the abstract "Little was known..." is not clear

- Present and past tenses should be adjusted

- Introduction: the term "intimate connection" is used to describe relationships between people, other adjectives should be used to underline connection between scientific measures.

- Sentences beginning with the word "While" should be reconstructed to improve the clarity: lines: 55, 178, 208, 217, 266

Lines 91-93 - This part is no clear, because there is no sentence, lack of verb.

The use of plural form "researches" seems improper. Usually the single form "research" is used. Maybe "research studies" could substitute "researches"

Author Response

  1. English editing is required to make the text more comprehensible. Parts of the text that require special attention are the following:
  • First sentence of the abstract "Little was known..." is not clear
  • Present and past tenses should be adjusted
  • Introduction: the term "intimate connection" is used to describe relationships between people, other adjectives should be used to underline connection between scientific measures.
  • Sentences beginning with the word "While" should be reconstructed to improve the clarity: lines: 55, 178, 208, 217, 266
  • Lines 91-93 - This part is no clear, because there is no sentence, lack of verb.
  • The use of plural form "researches" seems improper. Usually the single form "research" is used. Maybe "research studies" could substitute "researches"

Responses: Thank you for your comments and suggestions. The above language problems were corrected in the revised manuscript.

1. Abstract - First sentence not clear (see language comments above)

Responses: Thank you for your suggestions. We had corrected the related contents in the revised manuscript as follows: “The associations of combination with dynapenia and abdominal obesity and transitions with frailty progression remain unclear among middle-aged and older adults.”

2. Introduction - Please check if citation [13] is adequately used. In the manuscript it relates to dynapenia and physical frailty while the title of the publication [13] is related to cognition.

Responses: Thank you for your suggestions. We had corrected the related citations in the revised manuscript to show that dynapenia, presented by condition of age-associated loss of skeletal muscle strength was a key contributor to physical frailty using the most common definition proposed by Fried, et al. The revised sentence was as follows: “Weakness, with underlying mechanisms of impairments in the structure and function of the nervous and muscular system, had been considered as a key contributor to physical frailty.”

3. Materials and methods: Clearly described. No comments except for English editing.

Responses: Thank you for your suggestions. We had paid attention to English editing in the materials and methods section of revised manuscript.

4. Results: Good presentation with a lot of supplementary materials with detailed results. One issue that requires clarification: Was low BMI analyzed as a possible frailty correlate? Loss of weight and low BMI are known risk factors for frailty. If it was not included in the study, it should be mentioned as a limitation of the research.

Responses: Thank you for your suggestions. Firstly, we used the 32-item frailty index as a comprehensive and effective health-deficits indicator for assessing the frailty status in the present study, rather than the Fried phenotypes including weight loss, exhaustion, weakness, slowness, and low physical activity. In addition, considering loss of weight or low BMI are known risk factors for frailty, we included BMI as an adjusted covariate when analyzing the association and thus explored the independent effect of dynapenic abdominal obesity phenotype on the frailty progression.

5. Discussion: This section is rather concise, but includes all relevant findings of the study. Expanded discussion on the potential application of the research in practice (clinical medical practice, public health strategies) would enrich the discussion.

Responses: Thank you for your suggestions. We had expanded the potential application of our research in the revised manuscript as follows: “Despite the conclusions in this study need to be taken with caution when exploring potential intervention strategy, multidisciplinary teams including healthcare providers, nutritionists, therapists, as well as policy makers should develop comprehensive strategies to prevent and manage dynapenia and abdominal obesity against frailty among older adults. Health education should be widely carried out in primary health units to improve the awareness of maintaining muscle strength and preventing abdominal obesity from middle age. Furthermore, public health programs including aerobic and resistance training, and nutritional supplements should be developed aiming at improving muscle function, reducing abdominal fat, and preventing frailty progression.”

6.  Conclusions are in line with the results.

Responses: Thank you for your suggestions. We had revised our conclusions in the revised manuscript.

7. References: Please, check [13] - is the citation appropriate in the Introduction section?

Responses: Thank you for your suggestions. We had corrected the related citations in the revised manuscript as follows: “ 13.Clark BC, Manini TM: What is dynapenia? Nutrition 2012, 28:495-503. 14.Fried LP, Tangen CM, Walston J, Newman AB, Hirsch C, Gottdiener J, et al. Frailty in older adults: evidence for a phenotype. The journals of gerontology Series A, Biological sciences and medical sciences. 2001;56(3):M146-56.”